# Peer review of "Rapid and Sensitive Detection of Sulfamethizole Using a Reusable Molecularly Imprinted Electrochemical Sensor"

_foods, 2023, doi:10.3390/foods12081693_

Round 1

Reviewer 1 Report

In this research, a label-free electrochemical sensor for the determination of sulfamethizole was developed based on an electropolymerized molecularly imprinted polymer (MIP) film in a glassy carbon electrode. The sulfamethizole (SMZ) is utilized as a model of sulfonamides. The study is interesting. It remains to scientifically discuss the results obtained. It is necessary to include electron microscopy studies (SEM or TEM) of MIP. Therefore, I recommend that a minor review be performed.

General comments:

1. Because the Foods is a periodic dedicated to food, it is necessary to add a discussion on the presence of SMZ in food in the introduction. What published studies have detected SMZ in food? and what foods?

2. Some published works have already reported the MIPs technology for determination of sulfamethizole: Sensors and Actuators B: Chemical, 320 (2020) 128600; International Journal of Electrochemical Science, 14 (2019) 11630 – 11640, and Sensors and Actuators B: Chemical, 255 (2018) 3374-3383. Thus, the novelty of the present manuscript has to be well-detailed in the introduction, citing what was done differently from these cited references.

3. It is necessary to include electron microscopy studies (SEM or TEM) of MIP.

4. Update the terminologies for the electrochemical methods according to new IUPAC recommendations. See the information in Pure and Applied Chemistry, 92 (2020) 641–694. Current is I (in italics).

5. The authors, like many others, confuse the terms "detection" and "determination". Detection is qualitative by nature, while determination always is quantitative. Qualitative analysis is the detection of the presence of ions or compounds in an unknown sample, for example. The term "determination" refers to quantitative analysis to obtain data on the amount of analyte by weight or by concentration of an element or a compound in a sample. Therefore, most of the words “detection" in the manuscript should be replaced by the term "determination" (or "quantitation" or "assay") if quantitative assays are involved.

Specific comments:

1. Introduction:

a. Lines 48-51. We commonly describe the negative features of chromatographic or atomic absorption spectrometry techniques to laud our (electroanalytical) technique. I believe that all analytical techniques have advantages and disadvantages and that they all have space and function in scientific research. Thus, we can describe the positive characteristics of electrochemical methods, without diminishing the other techniques. This is just an opinion. Authors do not need to answer or comment.

b. Lines 53. It is important to add the advantages of using electrochemical methods. The use of miniaturized and portable sensors is only possible because electrochemical instrumentation is also miniaturized and portable. So, add the sentence: “In that respect, electrochemical methods offer the benefits of elevated sensitivity, accuracy, convenient operation, cheap instrumentation, facile integration, and portability”. Add the references Chemosensors, 10 (2022) 357 or other related references.

c. Line 88. The correct is “glassy carbon electrode”.

2. Experimental:

a. Standardize the description of the equipment: model (company, country).

b. Line 145. The term "detection" refers to a qualitative assay. Thus, the term "quantitative detection" is incorrect. Replace with just "determination".

c. Line 155. Do not use the acronym PBS for "phosphate-buffered solution". PBS is the accepted acronym for "phosphate-buffered saline" which contains 0.9% NaCl to warrant physiological ionic strength. See the Sigma Aldrich catalog (product P5368), for example. Also, see https://en.wikipedia.org/wiki/Saline (medicine). Unfortunately, PBS is often wrongly used as an acronym for "phosphate buffer(ed) solution" in the literature but this is wrong and can cause confusion.  Does the buffer employed by the authors really contain 0.9% NaCl (or other electrolytes such as MgCl2)? If yes, please specify. See https://en.wikipedia.org/wiki/Phosphate-buffered_saline.

3. Results and discussion:

a. It is necessary to include the Randles circuit used to fit the Nyquist spectra.

b. The caption of Figure 4 should be more complete, with more experimental information.

c. Line 303. The term "detection" refers to a qualitative assay. Thus, the term "quantitative detection" is incorrect. Replace with just "determination".

d. Line 359. Replace "detection" with "determination".

e. Table 1. If no SMZ was found in the real samples (no spiked), add a line indicating that at 0, no SMZ was detected. Before the spikes.

Reviewer 2 Report

The idea of using the technique of molecular imprinting (MI) in biosensors and immunodetection is not new. Nevertheless, the authors have introduced a number of new elements in the execution. So, in order to achieve effective recognition, monomers were screened among four types of 3-substituted thiophenes by computer modeling and subsequent experimental evaluation, and finally 3-thiophenethanol was selected. The formation of the MIS itself was carried out electrochemically. The time of the detecting layer of MI was very fast and took 30 minutes.

The authors obtained a good linearity in the range of 0.001-10 mM for this compound. The registration parameters were satisfactory - a low detection limit of 0.18 - 23 nM was reached for sulfamethisole. The sensor showed excellent selectivity, which makes it possible to distinguish structurally similar SAs. Even after 7 days of storage or reuse 7 times, more than 90% of the original detection signals were preserved.

The study was done at the modern methodological level.

As a comment, it can be noted that practical tests would require expanding the geography of samples. In this regard, it would be useful to conduct a broader verification of the reliability of the method. In addition, some disadvantage of the work is the use of GCE as a sensor electrode.

It would be good if the authors will describe the methods applied to electrodes of the type obtained with the help of screen printing - SPE.
